# Association between Dietary Habits and Severity of Symptoms in Premenstrual Syndrome

**DOI:** 10.3390/ijerph20031717

**Published:** 2023-01-17

**Authors:** Cinzia Quaglia, Immacolata Cristina Nettore, Giuseppe Palatucci, Fabiana Franchini, Paola Ungaro, Annamaria Colao, Paolo Emidio Macchia

**Affiliations:** 1Dipartimento di Medicina Clinica e Chirurgia, Scuola di Medicina e Chirurgia, Università degli Studi di Napoli Federico II, 80131 Naples, Italy; 2Istituto per l’Endocrinologia e l’Oncologia Sperimentale (IEOS) “Gaetano Salvatore”, Consiglio Nazionale delle Ricerche, 80131 Naples, Italy; 3UNESCO Chair on Health Education and Sustainable Development, Università degli Studi di Napoli Federico II, 80131 Naples, Italy

**Keywords:** premenstrual syndrome, diet, diet composition, copper, prevention

## Abstract

Background. Premenstrual syndrome (PMS) is a set of physical, psychological, and emotional symptoms that occur during the luteal phase of the menstrual cycle. The etiopathogenesis of this condition is not fully understood, and several studies suggest a possible role of environmental factors, such as diet. The aim of this work was to investigate the relationship between dietary habits and the occurrence and severity of PMS. Methods and Results. Forty-seven women were enrolled in the study. Participants were asked to complete the Daily Record of Severity of Problems (DRSP) to diagnose PMS and to complete a three-day food record during the perimenstrual phase. Thirty women completed the study (16 with PMS and 14 controls). An analysis of the food diaries revealed no differences between the women with PMS and the control subjects in terms of total energy intake (1649 vs. 1570 kcal/day), diet composition, and the consumption of macro- or micronutrients, except for copper, whose consumption was higher in women with PMS than in the control subjects (1.27 ± 0.51 vs. 0.94 ± 0.49 mg/d, *p* < 0.05). Conclusions. The data presented here are very preliminary, and only a significant difference in copper intake was found when comparing women with PMS and controls. Larger studies are needed to better define how diet may contribute to the exacerbation of the psychological and somatic symptoms associated with PMS and whether PMS itself may influence macro- or micronutrient intake by changing dietary habits.

## 1. Introduction

During the luteal phase, the period before menstruation, about 80% of women of childbearing age suffer from a variety of somatic, emotional, and psychological symptoms. In some cases, the symptoms are severe enough to cause significant discomfort and affect quality of life by interfering with normal daily activities, work, school performance, or interpersonal relationships [1]. These cases are referred to as premenstrual disorders (PMD) and can be classified based on symptom profiles as (a) predominantly psychological PMD; (b) predominantly somatic PMD; and (c) mixed PMD [2]. According to the American College of Obstetricians and Gynecologists (AGOC), the presence of nonspecific symptoms that can be confined to the luteal phase and thus have a plausible association with hormonal changes in the menstrual cycle is pathognomonic for premenstrual syndrome (PMS) [3].

Premenstrual symptoms are very common and affect approximately half of women of childbearing age worldwide [4]. When looking at the prevalence of PMS in different geographic regions, a discrepancy in the data becomes apparent. In the United States, PMS affects between 20 and 30% of women [5], while the prevalence in France is approximately 12% [6] and it reaches 98% in Iran [7]. This variability could probably be due to different diagnostic criteria and testing procedures but could also be due to different genetic or environmental factors such as lifestyle habits or diet.

Although reliable data are not yet available, the etiology of PMD appears to be multifactorial. The exact role of hormones in the etiology of cycle symptoms is unclear [8]. It has been suggested that symptoms occur after the preovulatory peak of estradiol and the postovulatory surge of progesterone in women who are highly sensitive to physiological hormonal fluctuations [9]. However, this theory does not explain why symptoms begin at ovulation in some women and occur at the end of the luteal phase in others [10]. The importance of progesterone versus estrogen in triggering symptoms is not as clear. Progesterone appears to be the defendant, and progestinic treatment has been used in the treatment of PMS [11], but it cannot be ruled out that estradiol independently exacerbates progesterone-triggered dysphoria [12,13] or similar progesterone-triggered symptoms [14]. Gonadal hormones interact with many other hormonal systems and the secretion of neurotransmitters such as gamma-aminobutyric acid (GABA) and serotonin. For example, the release of serotonin can be decreased by the action of progesterone, which in turn increases monoamine oxidase (MAO). This modulates the availability of 5-hydroxytryptamine (5 HT), leading to depressed mood [15].

Obesity and metabolic syndrome are thought to promote the occurrence of PMS. Indeed, PMS seems to be particularly common in women with a BMI > 27.5 kg/m^2^. Women living with obesity (BMI ≥ 30 kg/m^2^) have an almost three-fold increased risk of PMS compared to nonobese women (OR = 2.8, 95% CI = 1.1, 7.2). PMS is likely caused by a complex interaction of hormonal and neurochemical factors, and obesity may increase risk through multiple mechanisms. Compared with women with a BMI < 20 kg/m^2^, women with a BMI ≥ 30 kg/m^2^ have 39% lower follicular estradiol, 20% lower luteal estradiol, and 20% lower progesterone levels. Estrogen enhances the effects of serotonin by increasing its synthesis, transport, reuptake, and receptor expression as well as its postsynaptic responsiveness. Therefore, it is plausible that the lower levels of estradiol associated with obesity may lead to impaired serotonin function and contribute to the occurrence of PMS [16,17,18]. Several reports have suggested that the presence and/or severity of PMS may be influenced by dietary habits and nutritional status [19,20]. PMS has been associated with alcohol [21] and the consumption of various macro- and micronutrients (total carbohydrates, total fat, saturated and polyunsaturated fatty acids, thiamin, riboflavin, vitamin B6, vitamin D, calcium, magnesium, sodium, potassium, and zinc) [22]. In addition, PMS itself can influence food choices, which in turn can exacerbate or alleviate symptoms, so PMS symptoms and food intake are closely linked and may influence each other [23].

The present study is a pilot study that was designed to investigate the association between PMS and diet in a small group of young women to determine if certain nutrients can modulate the prevalence of this syndrome.

## 2. Materials and Methods

### 2.1. Participants

Forty-seven volunteer women aged 19 to 49 years were enrolled in the study. The exclusion criteria were pregnancy, menopause, primary or secondary amenorrhea, alteration of thyroid function, or psychiatric illness. The study was approved by the local ethics committee and carried out in accordance with the Declaration of Helsinki for human experimentation. The aim of the study was clearly explained to all study participants, and written informed consent was obtained. After signing the informed consent, all participants were asked to complete a questionnaire after the collection of anthropometric data (weight and height), which included the following medical history information: date of birth, the presence of diseases and any pharmacological therapies, and physical activity. In addition, specific menstrual cycle information was recorded, including the age of menarche, regularity, and duration of menstrual cycles and the current or previous use of estrogen–progestin pills or other contraceptive methods. A copy of the questionnaire that was used is included as a Appendix A. The severity of PMS symptoms and life impairments at different phases of the menstrual cycle were assessed using the Daily Record of Severity of Problems (DRSP) form, a validated prospective questionnaire that was developed to assist clinicians in diagnosing PMS [24]. To this end, the participants were appropriately trained to obtain an accurate and reliable symptom assessment prior to completing the DRSP. The DRSP was completed twice by each participant, and a PMS diagnosis was confirmed if the recorded symptoms were present during the premenstrual luteal phase (the last 5 days before menstruation) and relatively absent during the follicular phase of the menstrual cycle (from day 6 to day 13 after the onset of menstruation).

### 2.2. Dietary Habits

During the luteal phase of the menstrual cycle, participants were asked to complete a 3-day food diary to collect information on eating habits and diet composition. The diary was completed on three (not necessarily consecutive) days, one of which was a holiday. As a reminder about the importance of completing the diary during the premenstrual phase, all participants received an email 10 days before the estimated day of menstruation. All participants were provided with descriptive information to correctly identify foods that were consumed and calculate portion sizes. All foods and beverages that were consumed (including dressings) were recorded. Portions were reported by household measurements (cups, spoons, etc.) or by weight, with as much detail as possible (e.g., preparation methods and brand names). A qualified dietitian reviewed the records with each participant to identify potential errors and missing information. To calculate energy, macronutrient, and micronutrient intakes, the collected diaries were processed using Metadieta software (METEDA, Ascoli Piceno, Italy).

### 2.3. Statistical Analysis

Statistical analyses were performed using GraphPad Prism software, version 9.4.0. The results were expressed as means ± standard deviations (SDs). Normal distributions were checked with the Shapiro-Wilk test. Differences between women with PMS and controls were compared with Student’s *t*-test for independent data for normally distributed variables and with a Mann–Whitney test in the case of non-normally distributed data. Nominal variables were compared with chi-square (χ^2^) or Fisher’s exact tests. Differences with *p* values less than 0.05 were considered statistically significant.

## 3. Results

### 3.1. Characteristics of Participants

Of the 47 women that were initially recruited, 30 (63.8%) completed the study. According to the DRSP, 16 women were diagnosed with PMS, whereas 14 women were not diagnosed with PMS (control group, CTRL) (Appendix A). The characteristics of the women in the two participating groups are shown in Table 1.

The mean BMI values were 23.25 ± 2.25 kg/m^2^ in women with PMS and 22.23 ± 2.10 kg/m^2^ in CTRL (no significant difference). The percentage of normal-weight women (BMI ≤ 24.9 kg/m^2^) was not significantly different between PMS (87.5%, 14/16) and controls (93%, 13/14).

The age of menarche, the interval between menstrual cycles, and the duration of menstruation did not differ between women with PMS and CTRL. Birth control pills were used by three (18.75%) of the PMS participants and one (7.14%) of the CTRL participants (*p* = not significant), while participating women did not use other contraceptive methods. Ten participants with PMS (63%) and six from CTRL (43%) reported regular physical activity (exercising at least three times per week for one hour).

### 3.2. Analysis of Food Diaries

The dietary intakes of PMS patients and CTRL subjects were compared by analyzing food diaries.

PMS patients had an average energy intake of 1649 ± 368 kcal, of which carbohydrates accounted for 46%, proteins accounted for 20%, and fats accounted for 33%. In CTRL subjects, the mean energy intake was 1570 ± 362 kcal, of which 44% was carbohydrates, 19% was protein, and 37% was fat (Table 2, Figure 1). No significant differences were found between the two groups. The diet of the women with PMS was balanced and in accordance with the intakes recommended by the Italian Society for Human Nutrition (SINU) for the Italian population. In contrast, the macronutrient distribution in CTRL subjects showed a slight increase in fat consumption and a decrease in carbohydrate consumption compared to the optimal distribution range suggested for the Italian population (carbohydrates = 45–60%; proteins 15–25%; and fats = 20–35%) [25].

In women with PMS, simple carbohydrates accounted for 12.82 ± 4.75% of total energy intake and fiber intake was 12.58 ± 4.22 g per 1000 kcal. In CTRL subjects, simple carbohydrates accounted for 13.99 ± 2.12% of total energy and fiber intake was 11.06 ± 2.97 g per 1000 kcal. The differences were not statistically significant (Table 3).

Table 4 shows, in detail, the lipid intakes of the PMS and CTRL groups. Cholesterol intakes were similar in the two studied groups (199.93 ± 96.78 vs. 184.97 ± 108.11 mg/day) and were in line with the recommendations [25]. In contrast, women in the PMS group consumed significantly less saturated fat than CTRL women (8.35 ± 1.80 vs. 10.51 ± 3.12 g/day, *p* = 0.026), whose intake was above the recommendations (<10% of total energy intake [25]). The estimated intake of omega-6 fatty acids corresponded to 3.10 ± 1.49% in women with PMS and 3.03 ± 1.07% of total kilocalories in CTRL women. The differences were not statistically significant, although these values were lower than the recommended intake for the Italian population (4–8% of total daily energy intake [25]). Omega-3 fatty acid intake was 0.59 ± 0.86% of total kilocalories in women with PMS and 0.44 ± 0.19% in CTRL subjects. The differences were not statistically significant, and only the CTRL women had a slightly lower intake than recommended (0.5–2% of total energy intake [25]). The ratios of omega-6 to omega-3 were similar in the two study groups (7.44 ± 2.53 in women with PMS; 7.13 ± 1.74 in CTRL women) and were in line with the SINU recommendations [25].

The mineral and vitamin intakes are shown in Table 5. The copper intake was significantly higher (*p* < 0.05) in women with PMS than in CTRL women, although both groups met the recommended intakes [25]. The consumption of other minerals was similar in PMS women and CTRL women, although both groups had inadequate intakes of Ca, Mg, Na, K Cl, Fe, Zn, Se, I, and Mn [25]. Since the zinc–copper ratio is associated with PMS [26,27], we also calculated this ratio and found a reduction in women with PMS (7.2 ± 4.3) compared to CTRL women (9.02 ± 3.2), although the difference was not statistically significant.

No differences were found in vitamin intake, but the estimated consumption of vitamins A, B3, B5, D, E, and K was inadequate in both the PMS and CTRL groups [25].

Tryptophan is a precursor of serotonin, and it has been suggested that lower intake of this element may be related to PMS [28,29]; the consumption of this amino acid was also calculated, but no significant differences were found between the PMS group and the CTRL group (Figure 2).

There were also no significant differences in daily coffee consumption between the women with PMS and the control group (PMS = 2.34 ± 1.32; CTRL = 1.93 ± 1.44 cups of coffee/day).

## 4. Discussion

This work was a preliminary study that was designed to investigate the relationship between dietary habits and the occurrence and severity of PMS. The results suggest that, at least in this small group of patients, there are no relevant differences in the intake of macro- and micronutrients between PMS patients and healthy controls, except for copper, whose intake was significantly higher in PMS patients.

The extent to which premenstrual disorders affect the health of women and their families is underestimated [30]. Therefore, it is extremely important to recognize this disorder, determine the factors responsible for the onset and severity of symptoms, and provide appropriate treatment.

To date, the etiopathogenesis of premenstrual disorders is not fully understood, but several studies have shown the reciprocal influence between PMS and environmental factors such as dietary habits [20,21,22]. The hormonal fluctuations that occur during the menstrual cycle are not only responsible for the regulation and onset of ovulation and bleeding but can also affect food intake, energy expenditure, and behavior. Estradiol has been shown to directly inhibit food intake [31]. In addition, energy intake has been found to be relatively constant during the follicular and ovulatory phases, which are characterized by higher estradiol levels, but tends to increase during the luteal phase, when progesterone plasma levels are elevated [31].

Previous studies have observed that women with moderate or severe PMS symptoms consume more calories and experience cravings more frequently during the premenstrual phase than during the follicular phase [16,17,18,32,33]. Johnson et al. reported that women with depressed mood tend to consume more carbohydrate-rich foods [34]. Along the same lines are the observations of Wurthman and coworkers, who found not only a tendency for increased total energy consumption in women with PMS during the luteal phase but also that PMS symptoms decreased in women who ate a high-carbohydrate, low-protein meal during the late luteal phase [35]. Furthermore, in the same paper, using the Hamilton Depression Scale, the authors showed that depression, anger, tension, fatigue, confusion, and sleepiness improved significantly after the meal. In contrast, eating a meal with similar properties during the follicular phase did not improve these symptoms [35]. In addition, increased cravings for certain sugary, fatty, and salty foods such as fast food, chocolate, pastries, and desserts have been observed in women suffering from PMS [36,37].

Based on these observations, we would have expected higher total energy, carbohydrate, and simple sugar intakes in women with PMS compared to CTRL women, but our results showed no differences between the two groups, which is consistent with the findings of Bryant and coworkers, who observed a modest but nonsignificant increase in energy intake in women with PMS [38].

It has been suggested that the protective effect of carbohydrates on PMS symptoms and mood is mediated by increased brain uptake of tryptophan and the subsequent synthesis of serotonin [39]. Decreased serotonergic function has been linked to the onset and progression of depression [40]. The majority of serotonin is synthesized by tryptophan [41], and therefore dietary tryptophan levels may influence PMS symptoms [28], as an acute deficiency of this amino acid has been shown to exacerbate PMS [29]. To investigate this aspect, tryptophan consumption was also estimated here. The intake of this amino acid was slightly lower in the PMS group than in the CTRL group, but the differences were not statistically significant.

Previously, it was shown that women with PMS consumed less fruit, vegetable protein (fruits, seeds, and legumes), seafood, and whole grains overall and more refined grain products, high-energy foods high in saturated fat, and simple sugars than women without PMS. This resulted in lower intakes of omega-3 fatty acids, vitamin E, B vitamins, magnesium, calcium, zinc, and isoflavones [42].

The consumption of fatty acids has been linked to some behavioral and mood disorders due to their anti-inflammatory effects [43]. We examined the levels of omega-3 and omega-6 fatty acids in our study groups. In women both with and without PMS, the estimated omega-6 intake was not sufficient to reach the reference levels recommended for the Italian population [25], while omega-3 intake did not reach the recommended levels only in the control subjects. Nevertheless, the optimal omega-6/omega-3 ratio was maintained in both groups. The essential fatty acid consumption was not statistically different between the PMS and CTRL groups.

No significant differences were found in micronutrients between the two study groups, although the intakes of several vitamins and minerals did not reach the recommended levels.

The analysis of the food diaries revealed a higher copper intake in women with PMS compared with CTRL women. Copper is a zinc antagonist, and increased copper intake may decrease the intestinal absorption of zinc [44]. A recent study that examined the roles of some micronutrients in the development of PMS, dysmenorrhea, and irritable bowel syndrome found that zinc and copper are involved in the development of these disorders, although their roles are still unclear [45]. It has been suggested that low zinc and high copper levels may modulate GABA synthesis or membrane transport, leading to a decrease in transmitter concentration in the synaptic cleft [46]. Depending on copper levels and individual susceptibility, these metal levels in the brain can promote depression, irritability, anxiety, nervousness, and learning and behavioral disorders [46]. In addition, elevated levels of copper and its transporter, ceruloplasmin, are associated with inhibition of the enzyme hydroxytryptophan decarboxylase, which decreases the production of the neurotransmitter serotonin [46]. Copper and zinc are essential elements that also play critical roles in the redox balance and in preventing oxidative damage to cells and tissues. Gold and colleagues reported that depressed mood; abdominal cramps; food-related pain; and increased appetite, weight, pain sensitivity, and breast sensitivity were associated with elevated levels of high-sensitivity C-reactive protein (hs-CRP), a biomarker of inflammation. Increased oxidative stress and decreased antioxidant capacity may therefore be involved in the exacerbation of PMS symptoms [26]. In addition, zinc deficiency has been found in women with PMS in the luteal phase [27], and it has been suggested that the relationship between zinc and copper intakes is more clinically relevant than the intakes of the individual minerals. Indeed, both Cu and Zn intakes in the PMS and control groups were in line with recommended nutritional values, but the Zn/Cu ratio was slightly reduced in PMS, although the differences were not statistically significant. Unfortunately, as indicated in the study limitations (see below), our data lacked the serum levels of these minerals, which would have been useful to clarify this critical aspect [47].

The dietary survey that was conducted also considered caffeine consumption. ACOG currently recommends that women with PMS avoid caffeine. This recommendation is based on the results of a few cross-sectional observational studies that have shown a strong positive association between caffeinated beverage consumption and the occurrence of PMS [48,49]. Other studies have found no association between coffee consumption and premenstrual syndrome [49], a finding consistent with the data in this paper. Women both with and without PMS who participated in our study consumed similar numbers of cups of coffee per day. Based on these data, we can hypothesize that there is no association between coffee consumption and the occurrence of premenstrual symptoms.

A final comment should be made regarding the adequacy of the diets of the women that were studied compared to the recommendations for the Italian population [25]. Although our study group reported in their diaries an energy intake equal to the estimated energy consumption (Appendix A) and showed a good distribution of macronutrients, the intakes of several micronutrients were not sufficient to meet the recommended requirements. Low intakes of various minerals (Ca, Mg, K, Cl, Fe, I, and Mn) and vitamins (A, B5, D, E, and K) were found in both groups. This may be due in part to the tendency of short-term diet diaries to underestimate micronutrient intakes but is consistent with previously reported data for both Italy [50] and the populations of other European countries [51].

Our study has some limitations, and we are aware that this is a preliminary study and further investigation is needed. The first limitation is the lack of comparison between the estimated consumption of the different nutrients and the biochemical values. Our results would have been more interesting if they had been supported by a comparison with the circulating levels of various minerals and vitamins. In contrast, the food diary is a validated method for estimating food intake, and we decided to use it for three days to obtain reliable results. However, it should be noted that a three-day food diary is very reliable for energy and macronutrient intake, whereas a more accurate assessment of micronutrient intake is only possible with longer diaries (at least 5–7 days) [52]. Second, the size of the studied population was very small. Unfortunately, less than 70% of the recruited women completed the study. This was expected since the study was voluntary and, despite frequent reminders from the authors to the participating women, only some of them completed the questionnaire. Had we used a seven-day diary, we would have expected an even larger number of subjects who did not complete the study. Nevertheless, this is a critical issue because many of the discrepancies we observed may not have reached statistical significance because of the small sample size. Finally, it should be noted that in our study about 30% of the participating women with PMS were dietetics or human nutrition students. In the CTRL group, only 16.7% had the same educational background. Therefore, we cannot exclude the possibility that cultural background influenced the dietary habits of the participants and the results of the food diary analysis. The total energy intake was consistent with the estimated energy consumption of the participating subjects (Appendix A), suggesting that no relevant misreporting of energy intake occurred in the diaries. It would have been useful to use a validation method to assess the reliability of the data collected by our participants.

## 5. Conclusions

In summary, the occurrence and recurrence of premenstrual symptoms is only a minor disturbance in some women, whereas in others they may be severe enough to affect quality of life. In most cases, permanent pharmacological treatment is not recommended, and alternative therapies should be considered. The present study investigated the possible role of environmental factors such as dietary habits in relation to their association with PMS.

The preliminary results of this study suggest that a significant difference in nutrient intake was found only for copper. However, more in-depth studies are needed to determine the true role of this element, and a larger study is needed to determine if additional differences in dietary habits can be detected.

Although the results presented here are still very preliminary, we believe that encouraging women to improve the quality of their diet could reduce not only the severity of PMS symptoms but also the risk of developing PMS.

## Figures and Tables

**Figure 1 ijerph-20-01717-f001:**
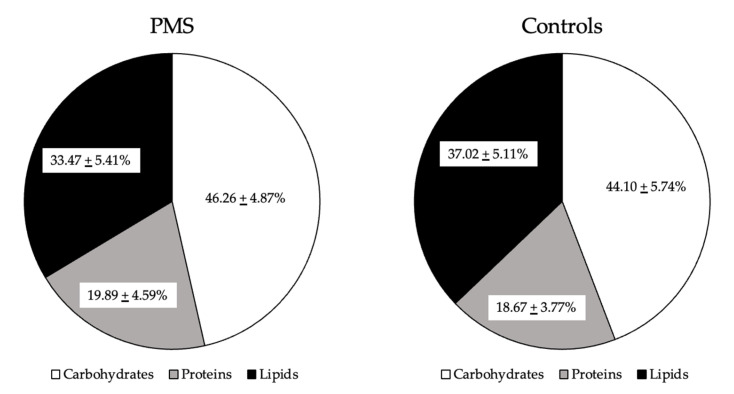
Macronutrient distribution ranges in the diets of participating women. The figure shows the estimated macronutrient distributions as percentages of total energy intake for the two study groups. The statistical differences were not significant. The macronutrient intake of PMS women followed the recommendation for the Italian population, while in the control group lipid consumption was slightly elevated and carbohydrate intake was slightly reduced compared to the optimal intakes (carbohydrates = 45–60%; proteins 15–25%; and lipids = 20–35%) [25].

**Figure 2 ijerph-20-01717-f002:**
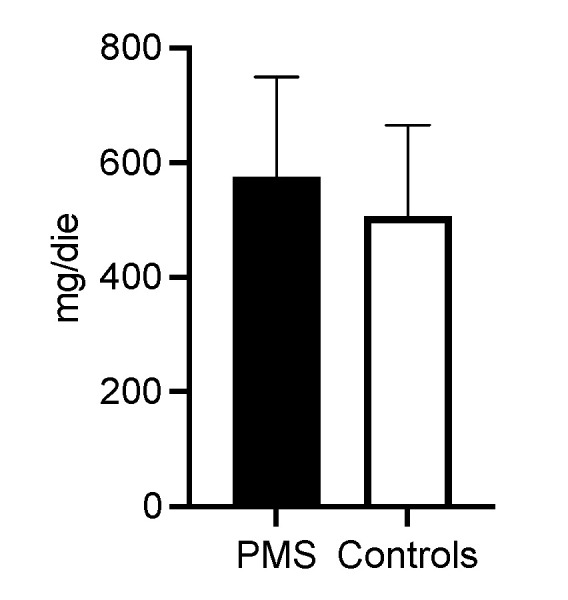
Estimated daily tryptophan intakes in participating women. Tryptophan intakes were calculated based on food diaries in both women with PMS and control women. This amino acid is a precursor of serotonin, and lower intake of this element has been associated with PMS [28,29]. No significant differences in tryptophan intake were found between the PMS group and the control group.

**Table 1 ijerph-20-01717-t001:** Characteristics of the studied population.

	PMS	Controls	*p* Value
Age	25.50 ± 2.32	28.79 ± 7.97	0.1392 ^a^
BMI (m/kg^2^)	23.25 ± 2.25	22.23 ± 2.10	0.2120 ^a^
Percentage of women with a BMI < 25	87.5% (14/16)	93.0% (13/14)	1.000
Percentage of women with a BMI < 30	12.5% (2/16)	7% (1/14)	1.000
Percentage of women with a BMI ≥ 30	0%	0%	1.000
Age of menarche	12.50 ± 1.5	11.71 ± 1.27	0.1498 ^a^
Average interval between menstrual cycles (days)	30.41 ± 4.06	27.71 ± 1.81	0.0799 ^b^
Average duration of cycles (days)	5.48 ± 0.93	5.0 ± 0.75	0.1520 ^a^
Percentage of participants taking birth control pills	18.75% (3/16)	7.14% (1/14)	0.6015
Regular physical activity	63% (10/16)	43% (6/14)	0.1679

^a^ Normally distributed values: *p* values were calculated with Student’s *t*-test. ^b^ Not normally distributed values: *p* values were calculated with Mann–Whitney test.

**Table 2 ijerph-20-01717-t002:** Estimated daily energy and macronutrient intakes in the participating women.

	PMS	Controls	*p* Value
Total energy intake (kilocalories)	1649 ± 368	1570 ± 362	0.5565 ^b^
Carbohydrates (g)	196.52 ± 48.66	181.14 ± 51.45	0.2400 ^a^
Proteins (g)	77.82 ± 19.63	70.68 ± 16.79	0.2969 ^a^
Lipids (g)	60.28 ± 20.05	63.35 ± 17.27	0.6591 ^a^

^a^ Normally distributed values: *p* values were calculated with Student’s *t*-test. ^b^ Not normally distributed values: *p* values were calculated with Mann–Whitney test.

**Table 3 ijerph-20-01717-t003:** Estimated daily carbohydrate intakes in the participating women.

	PMS	Controls	*p* Value	Recommended [25]
Total simple sugars				
g/day	54.70 ± 22.88	57.38 ± 16.71	0.7207 ^a^	<75
% of total Kcal	12.82 ± 4.75	13.99 ± 2.12	0.4037 ^a^	<15
Fibers				
g/day	20.68 ± 7.89	17.53 ± 7.087	0.2939 ^b^	≥17
g/1000 Kcal	12.58 ± 4.22	11.06 ± 2.97	0.3494 ^b^	≥8.7

^a^ Normally distributed values: *p* values were calculated with Student’s *t*-test. ^b^ Not normally distributed values: *p* values were calculated with Mann–Whitney test.

**Table 4 ijerph-20-01717-t004:** Estimated daily lipid intakes in participating women.

	PMS	Controls	*p* Value	Recommended [25]
Cholesterol (mg/day)	199.93 ± 96.78	184.97 ± 108.11	0.6921 ^a^	<300
Saturated fats				
(g/day)	15.39 ± 4.72	18.98 ± 8.94	0.4659 ^b^	
(% of total Kcal)	8.35 ± 1.80	10.51 ± 3.12	0.0684 ^b^	<10
ω6 (% of total Kcal)	3.10 ± 1.49	3.03 ± 1.07	0.9659 ^a^	4–8
ω3 (% of total Kcal)	0.59 ± 0.86	0.44 ± 0.19	0.5214 ^a^	0.5–2
ω6/ω3	7.44 ± 2.53	7.13 ± 1.74	0.3711 ^b^	4:1–8:1

^a^ Normally distributed values: *p* values were calculated with Student’s *t*-test. ^b^ Not normally distributed values: *p* values were calculated with Mann–Whitney test.

**Table 5 ijerph-20-01717-t005:** Estimated daily intakes of micronutrients in participating women.

	PMS	Controls	*p* Value	Recommended [25]
Ca (mg)	491.73 ± 190.46	495.60 ± 234.01	0.9606 ^a^	1000
P (mg)	886.84 ± 221.24	868.20 ± 287.48	0.8426 ^a^	700
Mg (mg)	198.86 ± 54.60	170.98 ± 49.22	0.1554 ^a^	240
Na (g)	1.15 ± 0.45	1.56 ± 0.76	0.0814 ^a^	<1.5
K (g)	2.27 ± 0.72	2.10 ± 0.59	0.4982 ^a^	3.9
Cl (g)	0.66 ± 0.54	0.87 ± 0.51	0.2196 ^b^	2.3
Fe (mg)	8.91 ± 2.35	9.03 ± 3.97	0.9185 ^a^	18
Zn (mg)	7.93 ± 2.50	7.61 ± 2.86	0.7499 ^a^	9
Cu (mg)	1.27 ± 0.51	0.94 ± 0.49	**0.0482 ^a^**	0.9
Zn/Cu	7.20 ± 4.35	9.02 ± 3.23	0.2102 ^a^	-
Se (µg)	36.56 ± 28.68	35.67 ± 27.85	0.9185 ^b^	55
I (µg)	53.12 ± 25.06	66.31 ± 49.98	0.7277 ^b^	150
Mn (mg)	0.93 ± 0.84	0.71 ± 0.44	0.6447 ^b^	2.3
Vit. A (µg)	146.47 ± 74.75	137.61 ± 76.15	0.7506 ^a^	700
Vit. B1 (mg)	0.84 ± 0.16	0.75 ± 0.33	0.3171 ^a^	1.10
Vit. B2 (mg)	1.19 ± 0.35	1.06 ± 0.36	0.3125 ^a^	1.10
Vit. B3 (mg)	15.22 ± 3.86	13.56 ± 5.52	0.3433 ^a^	14.0
Vit. B5 (mg)	2.38 ± 1.07	1.92 ± 0.66	0.1714 ^a^	5.0
Vit. B6 (mg)	1.50 ± 0.46	1.35 ± 0.48	0.3784 ^a^	1.3
Vit. B8 (µg)	18.42 ± 11.46	14.01 ± 7.85	0.1794 ^b^	5.0
Vit. B12 (µg)	2.95 ± 1.91	5.92 ± 8.37	0.7817 ^b^	2.4
Vit. C (mg)	106.26 ± 76.44	89.00 ± 48.64	0.6670 ^b^	75.0
Vit. D(µg)	2.18 ± 2.26	1.32 ± 1.20	0.0637 ^b^	100
Vit. E (mg)	9.38 ± 4.54	7.98 ± 2.40	0.4232 ^b^	15.0
Vit. K (µg)	8.56 ± 14.28	6.91 ± 6.8	0.5322 ^b^	140

^a^ Normally distributed values: *p* values were calculated with Student’s *t*-test. ^b^ Not normally distributed values: *p* values were calculated with Mann–Whitney test. The bold face indicates significant value.

## Data Availability

Not applicable.

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
