# Peer review of "Association between Dietary Habits and Severity of Symptoms in Premenstrual Syndrome"

_ijerph, 2023, doi:10.3390/ijerph20031717_

Round 1

Reviewer 1 Report (Previous Reviewer 2)

Thank you for setting out the changes to your manuscript. It is much improved and the limitations have been further highlighted. I have just a few further suggestions.

Abstract

Suggest re wording conclusion of abstract -

‘suggested that copper 25 intake may contribute to the exacerbation of psychological and somatic symptoms associated with 26 PMS.’ I don’t think you can make this statement on such as small sample and 3 day intake. You can say it is different and worthy of further research but not that it may contribute. It could be that PMS has made the participants change their diet in response to symptoms as you suggest in the introduction.

Introduction – new section in yellow/ line 66/67: it is thought best practice not to say ’obese women’ but ‘women living with obesity’. People first language rather than the disease being most important.

Results- carbohydrates spelt wrong on the pie charts Figure 1

Author Response

1) Summary. In accordance with the reviewer's suggestions, the abstract's conclusions have been changed.

2) Introduction, line 66. Thank you for pointing out this error. The text has now been changed in accordance with the reviewer's suggestion.

3) Results - Figure 1: The spelling errors have been corrected.

This manuscript is a resubmission of an earlier submission. The following is a list of the peer review reports and author responses from that submission.

Round 1

Reviewer 1 Report

This cross-sectional observational pilot study explores the impact of dietary habits on severity of symptoms in Premenstrual Syndrome. This is an interesting and relevant area of research.

There is good consideration of various aspects of the diet in this pilot study. However, there is a discrepancy between detailed data/ results in tables and in-text. This aspect can benefit from further clarity. Additionally, the introduction and discussion section are largely descriptive. This can be addressed by further detailing and contextualising statements. Where health benefits are claimed or difference/ impact has been inferred, study type should be considered and recorded, and differences between groups quantified in more detail, as well as considered in the context of human diets and recommendations.

Further comments relating to specific sections can be found below: 

1.    Abstract and Introduction: 

·       Where differences / impact are claimed, it would be beneficial to include further information such as study type and size of impact (e.g. lines 64, 67 – 72)

·       Abstract highlights differences in copper as the only dietary difference between groups, whilst data presented in the tables demonstrate differences for saturated fat and vitamin C as well.

2.     Methodology:

·       Line 79 – More specificity needed in the description of “relevant endocrine disorders”

·       Were non-hormonal birth-control methods such as copper coils considered?

·       Line 110 –Data analysis and descriptive statistics is detailed on assumption of normally distributed data. However, there is no information provided on whether data was tested for normality of distribution, what test was used for this and the corresponding results. Further details needed in this section.

·       Line 110 –Data analysis: What was alpha set at?

3.     Results:

·       Table 3: A significant p-value of 0.026 stated which does not correspond to the paragraph above

·       Table 5: The significant difference in vitamin C intake has not been discussed

·       Reason for drop-out rates need to be included

4.     Discussion

·       Aspects of the discussion does not correspond with the results. For example results demonstrate differences in saturated fat and vitamin C. However, the discussion does not consider these differences and highlights copper as the only element differing between the two groups.

Reviewer 2 Report

This is an interesting topic to investigate and the paper is well written.  I do have some concerns about the data collected and made comments below.

General comments:

title - it would sound better as 'dietary habits'

Abstract -It is better to write in the past tense - The aim of this work was to..

It is sounds more scientific to write 'energy' intake rather than calories or caloric intake.

The introduction is well written and informative.

Methods - I think a three day food intake is highly unlikely to reliably estimate micronutrient intake and to be able to estimate differences between the groups with sufficient accuracy. The food diary has limitations if food was not weighed and I think there is too much error over this short time. The sample population was small, there were lots of drop outs and the sample was not from the general population. This was all discussed in the limitations section but I think considering all of this the data is not sufficiently robust to make any conclusions.  

The dietary data needed to be validated in your population. At the very least there needed to be an estimate of under-reporting as the energy intake (and come micronutrient intakes - Ca, Fe, Vit A) were exceptionally low. This suggests significant under-reporting, which means the dietary data is not reliable enough to make any conclusions. A comparison of energy intake to energy expenditure or BMR can be used to estimate under-reporting. It looks to me like the dietary data is not complete. 

Statistics  - did you make any assessment of the normality of the data before using t-tests? All P values should be stated not just those significant. 

I noticed that some participants were taking birth control pills - is this not a contra-indication to taking part?

On page 4 - Table 3 and surrounding text - Is it total sugars or added sugars? Please add P values. 

Discussion - Start by summarising your findings and then compare to literature. Tryptophan and copper were adequate in the diets of your sample and so are not likely to be deficient and therefore be affecting PMS.

Line 239 - what are oily fruits?

I disagree that 3 days is sufficient to obtain reliable dietary results. In my view due to the limitations to the data collected and the small sample size, I think the conclusion is not supported y the data.